# The Male-Biased Expression of miR-2954 Is Involved in the Male Pathway of Chicken Sex Differentiation

**DOI:** 10.3390/cells12010004

**Published:** 2022-12-20

**Authors:** Yu Cheng, Zhen Zhang, Guixin Zhang, Ligen Chen, Cuiping Zeng, Xiaoli Liu, Yanping Feng

**Affiliations:** Key Laboratory of Agricultural Animal Genetics, Breeding and Reproduction of Ministry of Education, College of Animal Science and Technology and College of Veterinary Medicine, Huazhong Agricultural University, Wuhan 430070, China

**Keywords:** chicken, miR-2954, gonadal differentiation, vivo-morpholino, aromatase inhibitor

## Abstract

Many expression data showed miRNAs have a potential function on regulating gonadal differentiation in animals, but their function is rarely studied in vivo, especially in chickens. Using the comprehensive expression profiles analysis, the specific male-biased miR-2954, which is significantly higher expressed in male embryos and gonads at all detected stages, was firstly screened during the early stages of chicken embryogenesis and gonadogenesis. In sex-reversed female gonads treated with aromatase inhibitors, the expression of miR-2954 was increased, which was consistent with the up-regulation of *DMRT1* and *SOX9*. The injection of vivo-morpholino of miR-2954 significantly inhibited the expression of miR-2954 in chicken embryos, and the down-regulation of miR-2954 decreased the expression of testis-associated genes *DMRT1* and *SOX9*, while the expression of ovary-associated genes and the gonadal morphology did not change obviously. These results confirm that miR-2954 coincides with testicular differentiation in chicken embryos, but whether it might be an upstream cell autonomous factor to sex development in birds still need to be further determined.

## 1. Introduction

Sex development includes two processes of sex determination and sex differentiation. Sex determination in vertebrates is mainly influenced by heredity and/or environment. In mammals and birds, there is a well-known mechanism that sex is genetically determined by their sex chromosomes, on which the master trigger means sex-determining gene. The mammalian sex-determining gene is *SRY* (sex-determining region Y), which induces male pathway, so that individuals with the *SRY* gene develop into males, otherwise they develop into females [1]. Unlike mammals, birds have a unique ZW chromosome system in which males are the homogametic sex (ZZ) and females are heterogametic (ZW). Some important regulatory molecules which conserved among vertebrate species, such as *DMRT1*, *SOX9*, *AMH*, *FOXL2*, etc., have been identified during the process of gonadal sex differentiation, but the master sex determinant in birds remains unknown.

The classic model of sex determination in mammals states that the sex of the individual is determined by the type of gonad that develops, after which sex differences outside of the gonads are determined by differential effects of the gonadal hormones [2,3]. Other cell-autonomous sex chromosome effects, such as sex differences in body weight, adiposity and metabolic disease, are detected in mice in many phenotypes [3]. Birds show more widespread cell-autonomous sex determination in non-gonadal tissues because of ineffective sex chromosome dosage compensation mechanisms. In birds, analysis of several gynandromorphic chickens, which are rare bilateral sex chimeras, showed the birds had a mixture of ZZ and ZW cells that correlated with the asymmetric sexual phenotype, with ZZ predominating on one side and ZW on the other [4,5]. The sexually dimorphic structure of such gynandromorphic birds cannot be explained by sex steroid effects, which are expected to flow equally to both sides of the body. Arnold et al. [2,3] proposed a revised model of sex determination that recognized multiple primary parallel-acting factors encoded by sex chromosomes, which activate secondary downstream pathways (hormonal and genetic) to cause or reduce sex differences in phenotype. Much evidence points that the Z-linked *DMRT1* (Doublesex and Mab-3–Related Transcription factor 1) gene is the essential molecular trigger for sexual differentiation of testis development. The recent study of *DMRT1* mutated birds by using a CRISPR-Cas9-based monoallelic targeting approach showed that the chromosomally male (ZZ) chicken with a single functional copy of *DMRT1* developed ovaries in place of testes, but their appearance was indistinguishable to wild-type adult males, supporting the concept of cell-autonomous sex identity (CASI) in birds and indicating gonadal sex does not determine adult secondary sex characteristics [6].

In addition to protein-coding genes, more and more studies have shown that miRNAs, which are a kind of endogenous small RNAs with a length of about 20–24 nucleotides, also play key regulatory roles in animal gonadal development. In mice, several miRNAs were identified to be sexually dimorphic expression patterns at the time of sex determination and during the early fetal differentiation of gonads, among which miR-140-5p/140-3p has been proved to modulate Leydig cell numbers in the developing mouse testis [7]. Results also found the gene expression networks underlying fetal gonadal development are regulated by miRNAs in cattle and sheep [8,9]. In teleost, a number of sex-biased miRNAs have been identified during gonadal development of yellow catfish and zebrafish, and some of them are potentially involved in testis development and gametogenesis [10,11]. Similarly, there are plenty of sexually dimorphic miRNAs expressed during chicken embryonic gonadal development [12], such as miR-202 [13] and miR-363 [14].

MiR-2954, which is a sex-specific gene regulator encoded by Z chromosome in birds, was firstly reported in chickens to have an obviously higher expression in male embryos at an early developmental stage [4]. Studies in zebra finches showed that miR-2954 is broadly expressed across many tissues and highly expressed in all male tissues detected, such as auditory forebrain, heart, liver, gonad, and muscle [15,16]. Its function was suggested to mediate the neurogenomic effects of song response in the zebra finch [16,17]. MiR-2954 in chicken and zebra finches preferentially targets and down-regulates dosage-sensitive genes on the Z chromosome, which provides a new way to understand the expression mechanism of miRNA in sex dimorphism in birds [17,18]. The expression characteristics of miR-2954 suggest that it might be a cell autonomous factor to sex development in birds.

In this study, the male-biased expression of miR-2954 was firstly screened from chicken embryos by Solexa Deep Sequencing. Then, an aromatase inhibitor-treated and vivo-morpholinos-mediated chicken embryo model was used to study the expression changes of miR-2954 in gonads and its effects on sex differentiation-related genes and gonad development, so as to clarify the role of miR-2954 in sex differentiation and gonadal development of chicken.

## 2. Materials and Methods

### 2.1. Sample Collection

All animal experiment procedures were conducted following the guidelines for the care and use of experimental animals established by the Ministry of Agriculture and Rural Affairs of China. This project was also approved by the Scientific Ethics Committee of Huazhong Agricultural University (Wuhan, China). Fresh fertilized eggs of the breed Jingfen No.1 were obtained from Hubei Yukou Poultry Industry Co., Ltd. (Jingzhou, China). After being wiped with 75% alcohol, the eggs were incubated in the incubator under normal incubation conditions of 38 °C and 50–60% relative humidity until the desired embryonic stages. Chicken embryos at different stages were used for dissecting gonads or UGRs (urogenital ridge: gonad plus mesonephros) for RNA extraction, HE, fluorescence in situ hybridization (FISH), etc. The RNA-extracted gonads or UGRs were placed in RNase-free tubes and placed at −80 °C. The left gonads both in males and females used for immunofluorescence and HE were immediately embedded with a frozen section embedding agent and placed at −20 °C.

### 2.2. Embryo Sexing

Next, 1 μL blood collected from a chicken embryo was mixed with 9 μL 0.2 M NaOH, digested at 95 °C for 10 min, and then mixed with 9 μL 1 M Tris-HCl (pH = 8.0). Finally, 1 μL mixed sample or DNA extracted from other tissue or extra-embryonic membrane was used as the template for PCR sexing in a 10 μL reaction system including 5 μL 2 × M5 Hiper Plus Taq Hifi PCR Mix (AGbio, Changsha, China), 0.2 μL *CHD1*-F and *CHD1*-R (100 μM) primers, respectively. The PCR reaction conditions were 95 °C for 5 min, followed by 35 cycles at 95 °C for 30 s, 55 °C for 30 s, and 72 °C for 30 s, and extended at 72 °C for 5 min. The gender of chicken embryo was identified by the results of agarose gel electrophoresis.

### 2.3. Solexa Deep Sequencing and Data Analysis of MiRNAs

The whole embryos of E2.5 (embryonic day of 2.5) and E4.5, and gonads dissected from E5.5 embryos were collected and sexed, then pooled by sex with over 3 to 10 samples for each, named F25, M25, F45, M45, F55, and M55, respectively. Total RNA was prepared from pooled male and female gonad or embryo lysates using TRIzol (Invitrogen, Carlsbad, CA, USA) according to the manufacturer’s instructions. The construction of small RNA libraries and Solexa deep sequencing were carried out by BGI company (Shenzhen, China).

After evaluating the quality of sequencing data, all the low-quality reads, repeated reads, and adaptor sequences were removed from the raw reads, and the high-quality clean reads between 18 and 35 nucleotides (nt) in length were processed for further analysis. The sequencing results were mapped onto the G. gallus genome within Ensemble (http://www.ensembl.org/, accessed on 10 January 2020) using Bowtie and the Sanger miRBase (http://www.mirbase.org/, accessed on 10 January 2020) to identify known chicken miRNAs and conserved miRNAs. The RNA secondary structure of the remaining reads was further analyzed using miREvo [19] and mirdeep2 [20] to find novel miRNAs. The statistical significance of the different miRNA expression between female and male libraries at each stage was compared by calculating the log_2_ fold-change and *p*-value data of normalized expression.

### 2.4. In Vivo-Morpholino of miR-2954

In vivo-morpholino is a type of phosphorodiamidate morpholino oligomer improved by Gene Tool, LLC (Gene Tool, Corvallis, OR, USA). The sequence of in vivo-morpholino for anti-miR-2954 (VMO-miR-2954) was designed as 5′ AGCTCCACTACAATCCTCTCCCCAA 3′ and synthesized by Gene Tools (http://www.gene-tools.com/, accessed on 10 September 2020). The morpholino was resuspended in sterile phosphate-buffered saline (PBS) to form a 0.5-mM stock solution. Before microinjection, VMO-miR-2954 was prepared into 1 μg/μL with PBS. For the best systemic delivery results from blood vessels, the fertilized eggs were pre-incubated to embryonic day 3.0 and placed on a 2-cm square adhesive tape near the equator. A small hole was cut in the eggshell and 3 μL VMO-miR-2954 solution was injected into the major blood vessel using a Sutter-pulled glass capillary needle with the hand-held microsyringe under the stereomicroscope (MZ75, Leica, German). Eggs were sealed with clear tape and incubation was resumed. Blood was collected from yolk sac vessels at E6.0 for sex determination and then continued to hatch to desired stages.

### 2.5. Aromatase Inhibitor (AI) Treatments in Ovo

Fertilized eggs were treated with letrozole, a non-steroidal aromatase inhibitor (Catalog No. GC10726, GLPBIO). The letrozole was prepared to a 10 mg/mL stock solution in sterile PBS, and a single dose of 1 mg (100 μL) was injected into the albumin at the pointed end of each egg with a syringe at E3.0. Controls were injected with 100 μL PBS alone. Injected eggs were sealed with hot melt paraffin wax and re-incubated. The urogenital systems of embryos were collected at E10.0, and blood of each embryo was used for PCR sexing to identify the genotype of sex chromosome.

### 2.6. Quantitative Reverse Transcription-PCR (qRT-PCR)

Total RNA was extracted from embryonic tissues or gonads with TRIzol reagent (Invitrogen). After detecting the integrity and concentration, the cDNA was synthesized by using the PrimeScript RT reagent kit with gDNA Eraser (Takara, Beijing, China) according to the manufacturer’s protocol. The cDNA template for miR-2954 was prepared by stem-loop primers. Real-time PCR reactions for measuring the expression of sex-marked genes and miR-2954 were performed in the Bio-Rad CFX-96 and CFX-384 system using 2 × SYBR Green Pro Taq HS Premix II (AGbio, China) using *GAPDH* or 5S rRNA as an internal control (the primers are shown in Table 1). The final reaction volume of Q-PCR was 10 μL containing 5 μL SYBR Green Mix, 1 μL cDNA, 0.2 μL sense primer (10 mmol/L), 0.2 μL anti-sense primer (10 mmol/L), and 3.6 μL RNase-free ddH_2_O. The PCR procedure was 95 °C for 5 min, 40 cycles of 95 °C for 5 s, and 60 °C for 30 s. The relative expression of all genes used the comparative C_t_ method (ΔΔC_t_). Each group consisted of three biological repetitions, and all amplifications were performed with at least three technical repetitions.

### 2.7. Hematoxylin-Eosin (HE) Staining

The harvested embryonic urogenital systems were immediately frozen with liquid nitrogen after embedding with a frozen section embedding agent and placed into continuous sections under a tissue-freezing slicer (CM3050S, Leica, Wetzlar, Germany) with a thickness of 6 μm. Sections were fixed with methanol, rehydrated with ethanol, stained with hematoxylin-eosin, and dehydrated, and glass slides were inserted. Sections were visualized for histology under a microscope (BX53, Olympus, Tokyo, Japan).

### 2.8. Immunofluorescence Analysis

The embedded urogenital tissues were sliced continuously with the thickness of 10 μm in the frozen slicer. Then, the sections were fixed in 4% paraformaldehyde for 15 min, permeabilized in PBS containing 1% Triton X-100 at room temperature for 10 min, and blocked at 37 °C in PBS containing 5% BSA for 1 h. The primary antibody solution was diluted with 1% PBS, and DMRT1 antibody (1:100, AbClone, WG-03818, Wuhan, China), SOX9 antibody (1:100, Abcam, EPR12755, Cambridge, UK), FOXL2 antibody (1:200, Novus, NB100-1277s), and CYP19A1 antibody (1:50, Bio-Rad, MCA2077S) were added and incubated overnight at 4 °C. Secondary antibody goat anti-rabbit IgG (AbClone, AS007), goat anti-mouse (ProteinTech, SA00009-1, Wuhan, China), or rabbit anti-goat (ProteinTech, SA00009-4) was diluted with PBS at a ratio of 1:200, incubated at 37 °C for 1 h in dark, and washed 5 times with 1% PBS. Finally, sections were stained with 1 mg/mL of 4′, 6-diamidino-2-phenylindole (DAPI) for 10 min, sealed with an anti-fluorescence quenching agent, and imaged under a microscope equipped with fluorescence optics.

### 2.9. Fluorescence In Situ Hybridization

The biotinylated miR-2954 probes were synthesized by GenePharma Co., Ltd. (Shanghai, China). The miR-2954 probe: 5′ GUAGGGGUAAGG+UGAGGAUCGU-Biotin 3′; and the negative control probe (NC-Biotin): 5′ UGCUU+UGCACGGUAACGCC+UGUUUU-Biotin 3′. The miR-2954 FISH was performed according to the instruction of RNA FISH kit (SA-Biotin) (GenePharma). The slide preparation was performed as described above. The frozen sections were fixed with 4% paraformaldehyde for 10 min and washed twice with PBS. Sections were pretreated with Proteinase K (10 mg/mL) for 20 min. The miR-2954 probe was mixed with SA-CY3 at the ratio of 2:1 and used at 20 nM for incubation at 37 °C overnight. The slides were washed after hybridization and inoculated with DAPI. The images were obtained by a laser scanning confocal microscope (LSCM).

## 3. Results

### 3.1. Identification of Sexual Differentially Expressed miRNAs in Chicken Embryos and Gonads

To identify important miRNAs involved in chicken sex determination and sex differentiation, small RNA libraries between two sexes of E2.5, E4.5 whole embryos, and E5.5 gonads was constructed and sequenced using Solexa sequencing. After mapping and characterization for known miRNAs and novel miRNAs, differentially expressed miRNAs between females and males were screened by DESeq analysis (*q*-value < 0.01 and |log_2_ (foldchange)| > 1). As a result, 14 (up 6, down 8), 8 (up 4, down 4), and 19 (up 5, down 14) miRNAs were detected in the comparisons of F25 vs. M25, F45 vs. M45, and F55 vs. M55, respectively (Figure 1A–C). Multiple comparisons revealed only two miRNAs were found in all three comparisons, of which gga-miR-2954 was upregulated but gga-miR-6606-5p was downregulated in males at all three detected stages (Figure 1D,E). The details of differentially expressed miRNAs are shown in Appendix A. From the heatmap, four miRNAs (gga-miR-31-5p, gga-miR-2954, gga-miR-1456-5p, and gga-miR-2131-5p), which were located in the Z chromosome, were clustered with the similar profile of higher expression in males (Figure 1E).

### 3.2. The Spatiotemporal Expression Pattern of miR-2954 in Chicken Embryonic Gonads

The expression of four male-biased miRNAs on the Z chromosome in chicken embryos and embryonic gonads at E2.5–8.5 was measured by using qRT-PCR. Among these miRNAs, the significantly male-biased expression of miR-2954 was found at all investigated stages, which is similar to its RNA-Seq results (Figure 2D). However, the expression of miR-1456, miR-2131-5p, and miR-31-5p did not show obvious sex bias (Figure 2A–C). Further, the spatial expression of miR-2954 in E10 testis was analyzed by in situ hybridization. The results showed miR-2954 is highly expressed in seminiferous tubules of testicular medulla and is localized in the cytoplasm (Figure 2E–G).

### 3.3. MiR-2954 Expression in AI-Treated ZW Gonads

To prove the expression of miR-2954 associated with sex differentiation, sex-reversed ZW embryos were prepared by letrozole treatment at days of E3 and then sampled at E10. In the control group, bilateral testes were shown in males, which were cylindrical in size and tapering at both ends (Figure 3A), and asymmetrical ovaries developed in females with right gonad regression (Figure 3B). By contrast, the ZW gonads treated with AI were significantly different from the control female gonads and similar to the males’ testes in appearance with symmetrical distribution (Figure 3C). The control males exhibit typical testicular histological features with an obvious testicular cord in the medulla area and a thin cortex (Figure 3D). The ovary had a thicker cortical layer and vacuoles in the medulla (Figure 3E). In the masculinized gonads, the boundary between cortex and medulla was not obvious, the cortex became thinner, and the vacuolar structures in the medulla were reduced after AI treatment (Figure 3F).

Next, the expression of maker genes associated with sex differentiation was analyzed. Compared with the control group, the expression levels of *DMRT1*, *AMH,* and *SOX9* were significantly up-regulated (Figure 3G–I), while *CYP19A1* and *FOXL2* were significantly down-regulated in the AI female gonads (Figure 3J,K). Importantly, the expression level of miR-2954 in AI-treated female gonads was significantly increased when compared to control females (Figure 3L). In the immunofluorescence results, DMRT1 and SOX9 were specifically expressed in the gonadal medulla of males but not in females of control group (Figure 3M1,M2,N1,N2). After AI treatment, strong fluorescence signals of both DMRT1 and SOX9 appeared in female gonads (Figure 3M3,N3). Inversely, CYP19A1 and FOXL2 were only expressed in ovaries in the control group (Figure 3O2,P2), and they were significantly decreased in the AI-treated female gonads (Figure 3O3,P3). The fluorescence signal of miR-2954 was more intense in male testes than females by in situ hybridization, and it was also enhanced in AI-treated female ovaries (Figure 3Q1–Q3).

### 3.4. The Inhibitory Effect and Histomorphological Changes of Embryonic Gonads after VMO-miR-2954 Injection

The oligomer VMO-miR-2954 was transfected into chicken embryos at E3, then the expression changes of miR-2954 in chicken gonad, brain, and leg muscle were measured after treatment for 7, 9, and 15 days later, respectively. At the investigated stages of E10, E12, and E18, miR-2954 was highly expressed in male gonads, and it was significantly inhibited and similar to that in females after VMO-miR-2954 treatment (Figure 4A–C). The inhibitory effect of VMO-miR-2954 was gradually decreased with the development of chicken embryos, but the inhibitory level of miR-2954 was still less than half of that of the male control group until at E18. Furthermore, the expression of miR-2954 in brain, muscle, and gonad of male chicken embryos at E12 in MO treatment group was significantly downregulated when compared with the control group (Figure 4D), indicating that VMO-miR-2954 play a valid inhibitory effect in various tissues by the way of vascular injection in chicken embryos.

No obvious changes of gonadal morphology were observed with the naked eye after the inhibition of miR-2954, so HE staining was used to compare their histomorphology with control groups at E10, E12, and E18. In control groups, the gonads of male embryos have typical testicular characteristics with seminiferous cords in the medulla and a thinned cortex (Figure 4E,H,K), and the females show a gradually thickened cortical layer with a vacuolar structure in the medulla layer (Figure 4F,I,L). After VMO-miR-2954 treatment, the gonadal structure of male chicken embryos at each stage did not change significantly, but the outer cortex became thickened compared with the control group on the 18th day (Figure 4K,M). These results suggested that the inhibition of miR-2954 did not lead to significant changes in gonad morphology.

### 3.5. The Expression of Testicular and Ovarian Marker Genes after miR-2954 Inhibition

The expression of key markers involved in testis and ovary differentiation was examined after VMO-miR-2954 treatment at E10. The marker genes related to male sex differentiation are *DMRT1*, *SOX9,* and *AMH*, whereas key genes in female gonadal development are *CYP19A1* and *FOXL2*. At mRNA level, compared with the control male group, the expression of *DMRT1*, *SOX9,* and *AMH* in the male gonads of the treatment group were significantly down-regulated (Figure 5A–C), and *FOXL2* and *CYP19A1* expression did not change significantly (Figure 5D,E). The protein expression of four genes, DMRT1, SOX9, CYP19A1, and FOXL2, in embryonic gonads was detected by immunofluorescence. In the control groups, DMRT1 and SOX9 were specifically expressed in the male gonad and localized in the Sertoli cells of the medulla (Figure 5F,I), while CYP19A1 and FOXL2 were mainly expressed in the medulla of the female gonad (Figure 5M,P). After VMO-miR-2954 treatment, the expressed fluorescence of DMRT1 and SOX9 decreased in the male gonads (Figure 5H,K), and a small amount of CYP19A1 expression but no FOXL2 occurred in the medulla of male gonads (Figure 5N,Q), which is consistent with the RT-PCR results.

## 4. Discussion

As the special feature of taxonomic status, sex chromosome composition and gonad development among vertebrates, the molecular mechanism of sex determination, and sex differentiation in chicken have become the focus of research. Two Z-chromosome-linked genes, *DMRT1* and *cHEMGN*, have been demonstrated to be involved in chicken gonadal sex determination and sex differentiation. The knock-down of *DMRT1* in chicken reduced the expression of testicular marker *SOX9* and activated aromatase, while over-expression of *DMRT1* induced male pathway genes (*SOX9*, *AMH,* and *cHEMGN*) and antagonized female pathway genes (*FOXL2* and *CYP19A1*); these results indicated gonadal *DMRT1* plays a key role in testis development and support the Z dosage hypothesis for avian sex determination [21]. Chicken homolog of hemogen (*cHEMGN*) was expressed in the Sertoli cells of early embryonic gonads of male chickens after the sex determination and before the expression of *SOX9*, and it acts as a transcription factor in the molecular cascade between *DMRT1* and *SOX9* [22]. Here, we identified the unique small RNA miR-2954 is higher expressed in males from the early stages and found its expression was increased in AI-treated females. Based on the revised sex determination model suggested by Arnod et al. [2,3], miR-2954 encoded by the Z chromosome could be considered as a cell-autonomous sex determinant. Then, the miR-2954 morpholinos was transferred into chicken embryos to inhibit the expression of miR-2954 in males; with the expression of genes related to chicken sexual differentiation changed, these results suggested that miR-2954 may be involved in male pathway in chicken sex differentiation.

The male-biased expression of miR-2954 has been broadly studied in adult tissues of zebra finches. In chickens, Zhao et al. [4] firstly identified the obviously higher expression of miR-2954 in males in whole embryos at 48 h (H&H 14) and 72 h (H&H 20) of development before the sexual differentiation of the gonads, and this sexual dimorphic pattern also exist in adult tissues of lung, heart, liver, brain, gonads, breast muscle, kidney, and spleen. With the strong male bias, broad expression, and preference for Z-linked genes, miR-2954-3p might provide an alternative path to gene-specific dosage compensation [18]. In this study, we identified several differentially expressed miRNAs from the small RNA libraries between two sexes of whole embryos at E2.5, E4.5, and gonads at E5.5. Similar to the previous report of the small RNA sequencing data produced from chicken adult tissues, gga-miR-2954 was indicated to be the unique one which is consistently higher expressed in males across all detected stages of chicken embryos, which represents the stages before and after the sexual differentiation of gonads. Furthermore, miR-2954 was significantly male-biased in gonads at multiple stages of E5.5–8.5 and predominantly expressed in testicular medulla; we speculated that miR-2954 was an important regulatory factor in sex determination and the male developmental pathway in chickens.

During the gonadal differentiation pathway in birds, the bipotential gonads would develop into testes of males or ovaries of females. In males, bilateral testes develop a spermatic cord with cortical degeneration and medulla hyperplasia. In females, bilateral gonads develop asymmetrically, with gradual degeneration on the right side, thickening of the left cortex, and formation of a space in the medulla. Some reports have indicated that estrogen signaling plays an essential role in the differentiation of female ovaries in birds via estrogen receptor alpha (ERα) [23,24]. The suppression of estrogen levels in female embryos results in the disruption of the formation of gonads and the structure of ovary testosterone. Aromatase is the rate-limiting enzyme in the process of converting androgens to estradiol. Application of aromatase inhibitors can block estrogen synthesis in embryos. Injecting aromatase inhibitors into the eggs before the embryonic gonads began to differentiate, the female chicken embryos reversed their sex and formed bilateral testes with spermatogenic function [25]. In addition, the expression of *FOXL2* was decreased but not completely eliminated, while the expression of male genes *DMRT1*, *SOX9*, *AMH,* and *cHEMGN* were up-regulated in sex-reversal female embryos [22,26,27]. In our results, the female embryos in the AI-treated group showed significant changes of histological differentiation to the testis with the symmetrically distributed gonads and the presence of spermatic cord-like structures in the medulla. In the left female gonad treated with AI, the expression levels of the two female differentiation related genes *CYP19A1* and *FOXL2* were significantly decreased in both mRNA and protein levels, whereas the expression of the male genes *DMRT1* and *SOX9* began to appear. Our research indicates a degree of sex reversal in aromatase inhibitor-treated female embryos. In the sex-reversal chicken embryos, the expression of miR-2954 was up-regulated, which was consistent with the results of *DMRT1* and *SOX9* genes, indicating that miR-2954 is mainly involved in the process of testis differentiation.

Morpholino oligomer (MO) is an oligonucleotide composed of chains of about 25 subunits. The morpholino ring replaces the ribose ring for base complementary pairing. They bind to RNA to block translation, modify pre-mRNA splicing, and inhibit miRNA [28]. Morcos et al. [29] designed a membrane-penetrating transporter conjugated with morpholino to form vivo-morpholinos. In mice and zebrafish, target gene inhibition has been achieved by local or systemic intravenous administration [30,31,32]. In mice, low concentration of VMO can play an almost complete inhibitory role in tissues and organs through intravenous infusion [29]. Infusion of miR-34a-targeting VMO through the rat tail vein can significantly inhibit the expression of miR-34a in the liver [33]. In some cases, local injection can achieve better drug effects, and 2 days of intramuscular injection can significantly treat muscular dystrophy in mice [34]. So, that vascular injection of VMO-miR-2954 in chicken embryos is feasible, and VMO can effectively act on systemic tissues and organs compared with the delivery of MO by electroporation. We injected VMO-miR-2954 into the yolk sac vessel of the chicken embryo on day 3, which effectively inhibited the expression of miR-2954 in the gonad, even in brain and muscle. However, with the development of the embryo, the inhibitory effect of VMO-miR-2954 on miR-2954 expression was gradually weakened. In addition, Vivo-morpholinos can be used superimposed on the dosage to obtain a better knock-down effect [28]. In mice and zebrafish, the target gene is often significantly suppressed by injection through a tail vein, intraperitoneal injection, or direct local injection over a period of several days. However, the continuous injection is not easy to operate by the yolk sac vessel of chicken embryos.

In chicken embryos treated with VMO-miR-2954, the expression level of miR-2954 was significantly down-regulated in all tissues, and the expression of *DMRT1*, *SOX9,* and *AMH* were correspondingly reduced in male gonads. The above results suggest that miR-2954 may play a regulatory role in the upstream of *DMRT1* and *SOX9* genes and participate in the regulation of early sex differentiation and gonadal development in poultry. However, in terms of embryonic gonadal morphology, the inhibition of miR-2954 and the down-regulation of male marker genes were not enough to induce the obvious reversal of testis to ovary, indicating that miR-2954 perhaps does not play a key role in sex determination. Of course, the results of this study may also be related to the fact that the inhibition level of miR-2954 in chicken embryos is not sufficient.

From another fact of the molecular mechanism of miRNA, an miRNA can regulate the expression of multiple or even hundreds of target genes, and the target genes are also regulated by other miRNAs or transcript factors [35]. Some genetic disorders may not result in serious consequences and may themselves maintain homeostasis through other mechanisms. For example, miR-17-92 maintains spermatogenesis in mouse testes. When the sex of mouse embryos was determined, the mutant mice that knocked out miR-17-92 developed normal testes. This suggests that miR-17-92 contributes to the maintenance of normal gene expression levels but is not necessary for testicular development and function [36]. The knockout or overexpression of miR-2738 in Bombyx mori leads to changes in the expression level of downstream sex-determining genes, but does not lead to changes in the sex phenotype [37]. Therefore, miRNA may mediate a regulatory pathway that plays an auxiliary but not dominant role in regulating the expression of sex-determining genes.

Two miRNA clusters, miR-34a/b and miR-449a/b/c, co-regulate the spermatogenesis of mice, and the knockout of one of the miRNA clusters did not affect the reproductive ability of mice, while the mice were infertile after double knockout [38]. In poultry, a number of miRNAs have been identified that regulate gonadal sex differentiation, among which miR-31 is a significantly male-biased expression at E5.5 and E6.5 and predicted to target genes of the TGF-β signaling pathway, which plays a critical role during gonad development [39]. In addition, the expression pattern of miR-2954 was highly similar to that of miR-31 during the period of gonadal sex differentiation. We speculate that miR-2954 may co-act with miR-31 and other regulatory factors in the male pathway of sex differentiation. The inhibition of miR-2954 would lead to the dysregulation of downstream sex differentiation related genes, but the normal testicular structure was still developed, just indicating that the sex differentiation and gonadal development process in birds is a strict homeostasis process, and some of its functions may be compensated by other regulatory factors such as miR-31 after the inhibition of miR-2954.

## 5. Conclusions

In conclusion, our work further investigated the specific male-biased expression of miR-2954 during chicken early development of sex differentiation; the increasing expression of miR-2954 was found in masculinized ZW gonads, and its downregulation affected the expression of *DMRT1* and *SOX9,* which are involved in the male pathway. The comprehensive results indicated miR-2954 might be involved in the processes of testicular differentiation during chicken embryonic development. In further studies, the specific regulatory mechanisms of miR-2954 during sex determination and sex differentiation in poultry still need to be determined. 

## Figures and Tables

**Figure 1 cells-12-00004-f001:**
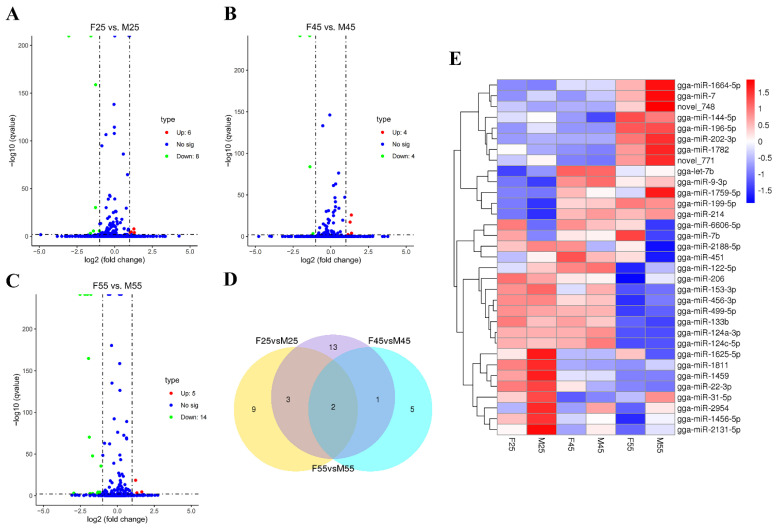
The differentially expressed miRNAs in chicken embryos and gonads. (**A**–**C**) Volcano plot of differentially expressed miRNAs in E2.5, E4.5, and E5.5 samples, respectively. The log_2_ (fold change) difference between females and males is shown on the *X*-axis, and the negative log of *q* values is shown on the *Y*-axis. Each point shows a gene with detectable expression in both samples. (**D**) Venn diagram of differentially expressed miRNAs. (**E**) Heat map visualization of the clusters of differentially expressed miRNAs between all samples. Values plotted are log_2_ fold change.

**Figure 2 cells-12-00004-f002:**
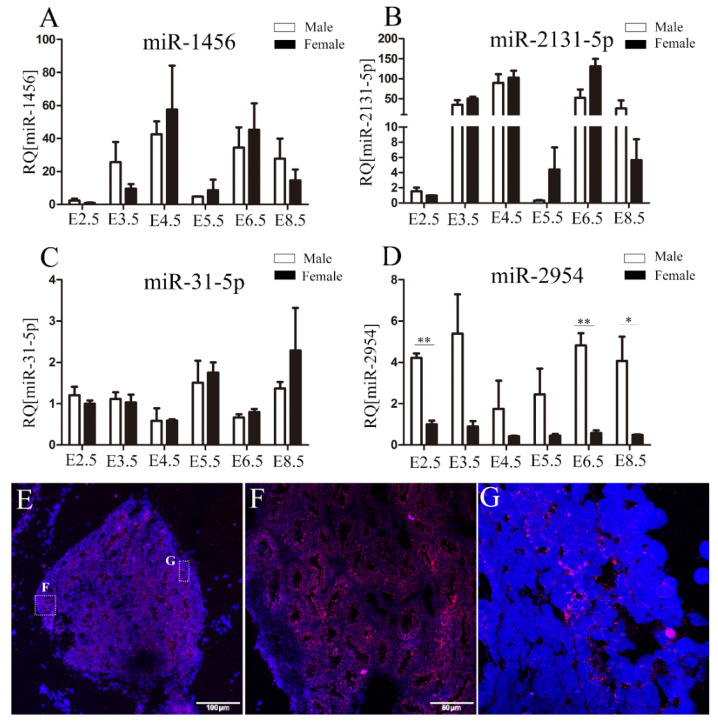
Validation of the expression profiles of miRNAs in chicken embryonic gonads. (**A**–**D**) The expression profiles of miR-1456, miR-2131-5p, miR-31-5p, and miR-2954 during E2.5–8.5, respectively. Each bar represents the mean and standard error of expression measured by qRT-PCR. Asterisks indicate statistical significance, * *p* < 0.05, ** *p* < 0.01. (**E**–**G**) The localization of miR-2954 expression in chicken embryonic testis at E10. DAPI staining of nuclei is blue and immunofluorescence of miR-2954 is red. E, scale bar = 100 μm; F, scale bar = 50 μm, thickness = 10 μm; G, magnification = 100.

**Figure 3 cells-12-00004-f003:**
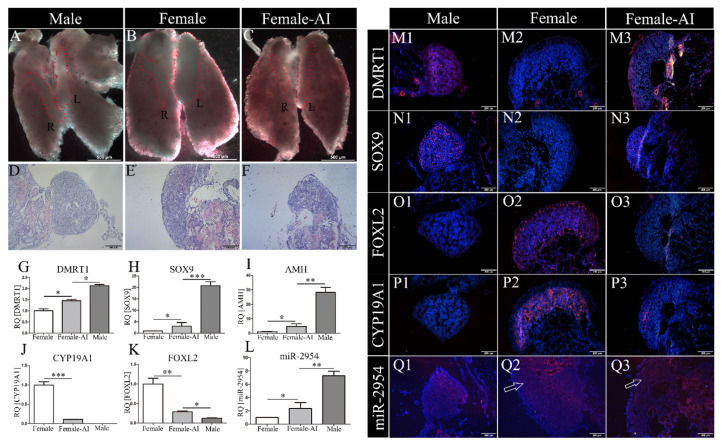
The morphology and gene expression in gonads at E10 chicken after AI treatment. (**A**–**C**) Gonads on top of the mesonephros in male, female, and AI-treated female chicken, respectively. (R) indicates right gonad, (L) indicates left gonad. (**D**–**F**) HE staining of gonads in male, female, and AI-treated female, respectively. In all HE staining images, the left gonad was selected. Scale bar as is shown in pictures, thickness = 6 μm. (**G**–**L**) Q-PCR expression of *DMRT1*, *SOX9*, *AMH*, *CYP19A1*, *FOXL2,* and miR-2954, respectively. Error bars show standard error of the mean (SEM) from at least three biological replicates in each group. Asterisks indicate statistical significance, * *p* < 0.05, ** *p* < 0.01, *** *p* < 0.0001. (**M1**–**M3**,**N1**–**N3**,**O1**–**O3**,**P1**–**P3**) Immunofluorescence results of DMRT1, SOX9, FOXL2, and CYP19A1 in control and AI-treated gonads. (**Q1**–**Q3**) In situ hybridization of miR-2954 in male, female, and AI-treated female. Blue fluorescence shows DAPI, which marks the nucleus; red fluorescence indicates expression of sex markers or miR-2954. Scale bar = 200 μm, thickness = 10 μm.

**Figure 4 cells-12-00004-f004:**
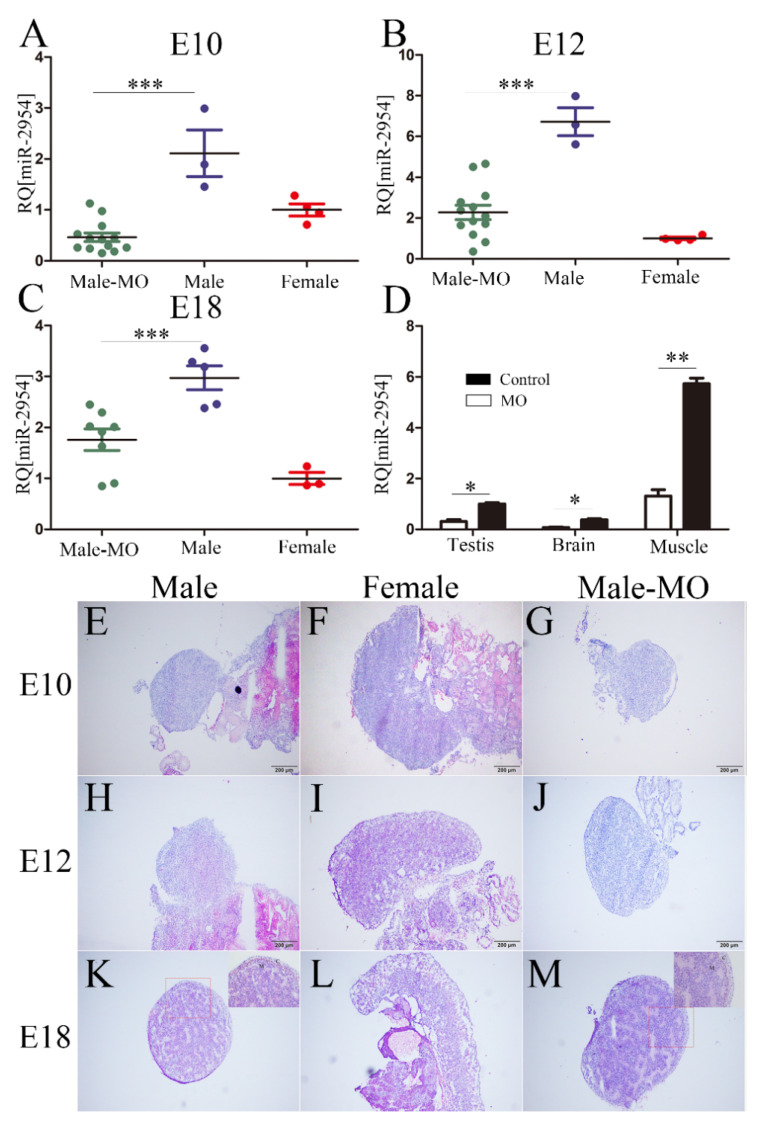
The expression of miR-2954 and HE staining of chicken embryonic gonads after VMO-miRNA-2954 treatment. (**A**–**C**) The expression of miR-2954 in chicken gonads after VMO-miR-2954 treatment at E10, E12, and E18, respectively. (**D**) The expression level of miR-2954 in different tissues of males at E12 after VMO-miR-2954 treatment. MO, VMO-miR-2954; M, male; F, female. Error bars show standard deviation (SEM). Asterisks indicate statistical significance: * *p* < 0.05, ** *p* < 0.01, *** *p* < 0.0001. (**E**–**G**) Embryonic gonadal histomorphology at E10 stage in male, female, and VMO-treated male, respectively. (**H**–**J**) Embryonic gonadal histomorphology at E12 stage in male, female, and VMO-treated male, respectively. (**K**–**M**) Embryonic gonadal histomorphology at E18 stage in male, female, and VMO-treated male, respectively. C, cortex; M. medulla. For all pictures, scale bar = 200 μm, thickness = 6 μm.

**Figure 5 cells-12-00004-f005:**
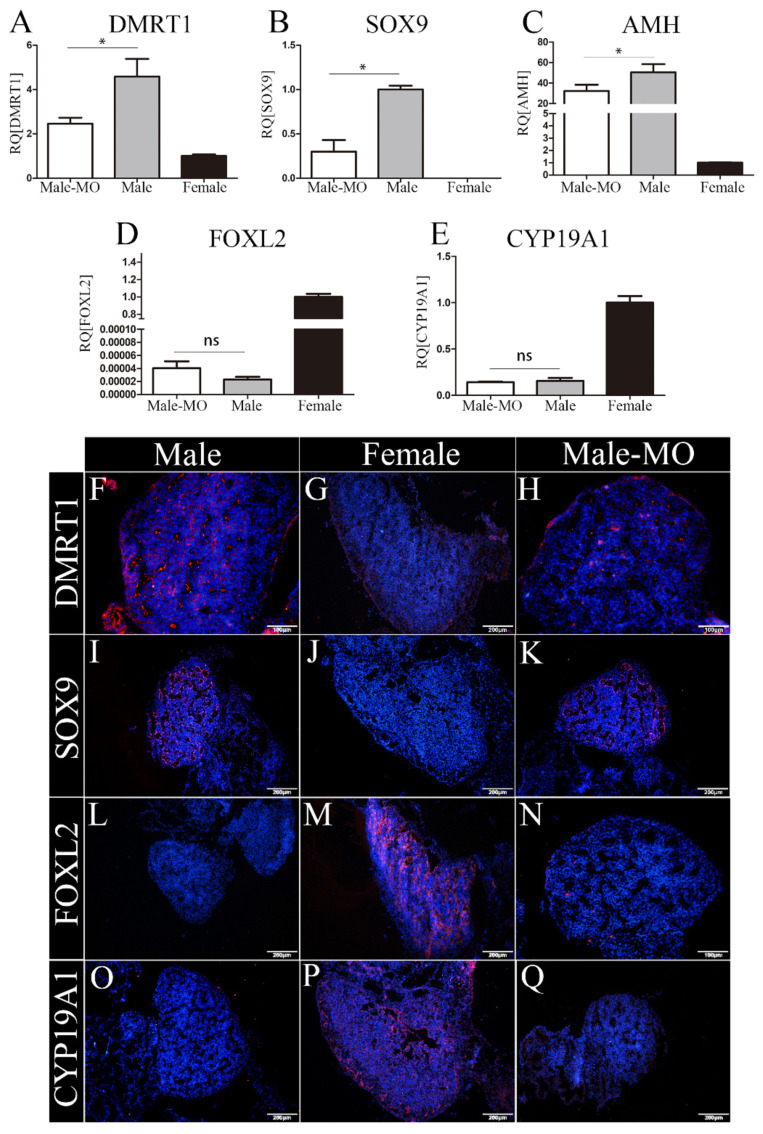
Expression of genes related to gonad differentiation at E10 after miR-2954 inhibition. (**A**–**E**), Q-PCR results of *DMRT1*, *SOX9*, *AMH*, *CYP19A1,* and *FOXL2*, respectively. Error bars show standard error of the mean (SEM) from at least three biological replicates in each group. Asterisks indicate statistical significance: * *p* < 0.05; ns means no significant difference. (**F**–**Q**) Immunofluorescence results of DMRT1, SOX9, FOXL2, and CYP19A1, respectively, in male, female, and VMO-treated male gonads. Blue fluorescence shows DAPI, which marks the nucleus; Red fluorescence indicates expression of sex markers. Scale bar as is shown in picture, thickness = 10 μm. M-MO indicates male chicken embryo treated by VMO-miR-2954.

**Table 1 cells-12-00004-t001:** Primers used in reverse transcription and Q-PCR.

Gene	Sequence (5′-3′)	Genebank No.
RT-miR-2954	CTCAACTGGTGTCGTGGAGTCGGCAATTCAGTTGAGTGCTAGGA	
RT-5S rRNA	AACTGGTGTCGTGGAGTCGGC	
gga-miR-2954-F	GGTAGGCATCCCCATTCCACTC	MI0013634
gga-miR-2954-R	AACTGGTGTCGTGGAGTCGGC	
F-5S rRNA	CCATACCACCCTGGAAACGC	M13919.1
R-5S rRNA	TACTAACCGAGCCCGACCCT	
*SOX9*-F	TTCTCGCTCTCATTCAGCAG	NC_052549
*SOX9*-R	GTACCCGCATCTGCACAAC	
*AMH*-F	CTCCCTCACCAACTACTCAACC	NC_052559
*AMH*-R	TGCCAGTCCCCAAAATGCT	
*FOXL2*-F	AGAACAGCATCCGCCACAA	NC_052540
*FOXL2*-R	GGGTCCAGCGTCCAGTAGT	
*CYP19A1*-F	GCTTGGATTACAGTGCATTG	NC_052541
*CYP19A1*-R	CCAGGACCAGACAGGGCT	
*GAPDH*-F	GAGGGTAGTGAAGGCTGCTG	NC_052532
*GAPDH*-R	CACAACACGGTTGCTGTATC	
*CHD1*-R	GTTACTGATTCGTCTACGAGA	NC_052572
*CHD1*-F	ATTGAAATGATCCAGTGCTTG	

## Data Availability

Data available on request from the authors.

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
