# Peer review of "The Male-Biased Expression of miR-2954 Is Involved in the Male Pathway of Chicken Sex Differentiation"

_cells, 2022, doi:10.3390/cells12010004_

Round 1

Reviewer 1 Report

In manuscript, Cheng et al found miR-2954 significantly upregulate in male embryos and gonads during chicken embryo early development through small RNA sequencing. Then they tried to uncover the correlation between miR-2594 and sex determination/ differentiation in chicken. miR-2594 previously has been reported to highly express in male embryos in chicken at early developmental stage. Therefore, although the authors use different methods to demonstrate the conclusion, it’s lack of novelty. As to the mechanism, the authors overall failed to explain how the miR-2954 affect sex determination/ differentiation in chicken during early embryo development. The following are some major and minor points need to be addressed.

Major points:

1.     What’s the phenotype of chicken embryos when overexpressing miR-2945?

2.     What’s the role of miR-2945 in the male pathway of chicken sex differentiation?

3.     In figure 6A-E, statistical analysis should be shown in each panel. In figure legend, it is said that statistical significance shown by letters, it is confused.

Minor points:

1.     In figure 2E-G, fluorescence should be annotated.

2.     I suggest combine 3.4 and 3.5.

Reviewer 2 Report

Dear Authors,

The Manuscript is well prepared and the molecular analyses and results obtained are very interesting. However, many key detailes nead correction.

Reviewere’s comments:

1.       Use Italic font for all Latin terms. Like in vivo, in vitro. (Line 2 of Abstract and others)

2.       Sample collection: For me it is not clear if you focused on the side of female gonad prepared to analyses. Left gonad develop, and right regress, and the expression of aromatase and other enzymes is different in left and right gonad in female birds. So, the way the sample was taken plays a key role. You wrote: Chicken embryos at different stages were used for dissecting gonads or UGRs (urogenital ridge: gonad plus mesonephros) for RNA extraction, HE…. Please, write which gonad was analysed. And write information about right gonad regression process.

3.       Ad 2.4. In Vivo-Morpholino of miR-2954

„A small hole was cut in the eggshell and 3 μL VMO-miR-2954 solution was injected into the major blood vessel using a Sutter-pulled glass capillary needle with the hand-held microsyringe under the stereomicroscope (MZ75, Leica, German). Eggs were sealed with clear tape and incubation was resumed. Blood was collected from yolk sac vessels at E6.0 for sex determination and then continued to hatch to desired stages.” From my experience, it is very complicated procedure and additionally day 3 of embryogenesis is a day, when embryo is extremally sensitive. What was the percentage of death embryos after injection of this liquid?

4.       Ad 2.4. Aromatase Inhibitor (AI) Treatments In Ovo

You have used 100ul of solution? It is very high volume. What was the percentage of death embryos after this procedure?

5.       Why you used tape in one procedure and parafin wax for another?

6.       Gene primers: Please, add information what was the GeneBank numer for each sequence

7.       Ad Immunofluorescence: Please, write about specifity of antibody for birds. What is the homology? Is is antibody against chicken proteins? If not, you have to test it using Western Blot and additionally compare aminoacids sequence.

8.       Figure 2:

You should use the same font for each figures. You can write „RQ [miR 14-56]” instead of „Relative expression of….” on the Graphs.

Pictures quality is very bad, are you sure if it is gonad? Not mesonephros?  I know, that it is very early stage of development, but try to correct the quality. And you have to present negative control without primary antibody.

Figure 3: G-L too small fond, use the same size and type of font everywhere

Figure 5: The quality of pictures is very bad.

Round 2

Reviewer 1 Report

The authors addressed my questions.

Reviewer 2 Report

In my opinion corrected version of this Manuscript may be accepted by the Editor.